# The Use of RE-AIM to Evaluate a Pharmacist-Led Transitions of Care Service for Multivisit Patients at a Regional Hospital

**DOI:** 10.3390/pharmacy13040099

**Published:** 2025-07-23

**Authors:** Courtney E. Gamston, Salisa C. Westrick, Mafe Zmajevac, Jingjing Qian, Greg Peden, Dillon Hagan, Kimberly Braxton Lloyd

**Affiliations:** 1Harrison College of Pharmacy, Auburn University, Auburn, AL 36849, USA; westrsc@auburn.edu (S.C.W.); mzr0058@auburn.edu (M.Z.); jzq0004@auburn.edu (J.Q.); pedengc@auburn.edu (G.P.); lloydkb@auburn.edu (K.B.L.); 2Intermountain Health, Grand Junction, CO 81501, USA

**Keywords:** pharmacists, transitional care, implementation science, quality improvement

## Abstract

Pharmacist-led transitions of care (TOC) services decrease preventable hospital readmission. TOC service implementation assessment can inform translation to real-world settings. The purpose of this study was to evaluate the implementation of a TOC service for patients with multiple admissions at a regional hospital using the RE-AIM framework. In this quasi-experimental, non-randomized study, individuals with ≥2 recent hospitalizations received pharmacist-led discharge medication reconciliation and counseling, management of drug-related problems, post-discharge telephonic visits, and social support. The reach, effectiveness, implementation, and maintenance RE-AIM dimensions were assessed using patient and service records. Outcomes included 30-day readmission rates for individuals completing ≥1 outpatient pharmacist visit (intervention) versus those unreachable in the outpatient setting (comparison), completed interventions, implementation features, and service adaptations. Chi-square and Fisher’s exact tests were used for comparison of categorical variables and the t-test was used for continuous variables. From February 2022 to August 2023, 72.7% of the 66 service participants participated in the intervention (reach). Additionally, 30-day readmission was 22.9% (intervention) versus 55.6% (comparison; *p* = 0.01). In total, 2279 interventions were documented (effectiveness). The service was adapted (implementation) and expanded to include additional populations (maintenance) to enhance sustainability. Based on RE-AIM evaluation, the pharmacist-led TOC intervention appears to be a sustainable solution for addressing readmission in multivisit patients.

## 1. Introduction

Transitions of care (TOC) occur as individuals move between levels of medical care, such as moving from the inpatient to outpatient setting [1]. Poor transitions from the hospital to community setting often result in early hospital readmission and increased morbidity and mortality [2]. For the top 18 diagnoses in the U.S. in 2020, the 30-day readmission rate ranged from 3.6% to 23.8% depending on admission diagnosis [3]. This variability in readmission risk between disease states has impacted reimbursement for hospital services under the Center for Medicare & Medicaid Services (CMS) Hospital Readmissions Reduction Program (HRRP). This program, established under the Affordable Care Act, utilizes a negative incentive structure based on readmission data for six high-risk conditions to improve care transitions. High relative readmission rates inform a penalty of 0–3% of the reimbursement for all Medicare admissions within the penalty period [4]. Service evaluations in the literature have tended to focus on impacts on participants with specific high-risk disease states [5,6,7,8], most commonly those that are the focus of the HRRP.

Repeat hospital admission is a risk factor for early readmission [9] and is associated with a patient’s medical care and additional contributing factors such as negative social determinants of health (SDOH), or social risks [10,11,12]. In a survey of elderly patients with a history of four or more annual admissions, common reasons for readmission included a lack of care coordination, poor communication, drug-related problems (DRPs), and poor social support, [10] all of which reflect recommended areas of focus for TOC services in the literature [13]. Further support for targeting frequently admitted patients is provided by the finding that this population has an increased mortality risk that persists for multiple years after discharge [14]. Given this, a recent history of repeat admission may be an effective criterium for identifying patients at high risk for readmission. Pharmacists are well positioned to provide TOC services to this high-risk population due to their proven effectiveness in medication management, patient and provider communication and education, the patient-centered care approach, and collaboration with other providers [15].

Evidence from numerous studies demonstrates that TOC programs that capitalize on pharmacist expertise reduce the risk for early hospital readmission by identifying and addressing DRPs and improving communication between patients and providers [15,16,17]. Studies of pharmacy-supported TOC programs have shown that readmissions can consistently be decreased by approximately one-third, demonstrating a significant impact on the risk for patient harm [18]. The purpose of this study was to use the RE-AIM implementation science framework [19] to assess the implementation of a pharmacist-led, ambulatory care-based, telephonic TOC service at a regional hospital that targeted patients with multiple recent hospital admissions. The RE-AIM framework can be utilized to evaluate the impact and implementation of healthcare interventions and guide the translation of research into clinical practice. Through the five domains (Reach, Effectiveness, Adoption, Implementation, and Maintenance), this framework provides structure to the evaluation of program components and elucidation of factors that contribute to successful program implementation [19]. The overall goal was to implement a service model that functioned within the context of the hospital workflow that consistently improved the rate of readmission for frequently admitting patients.

## 2. Materials and Methods

### 2.1. Design

The project was prospectively designed to evaluate the implementation and evaluation of a pharmacist-led, ambulatory care-based TOC service utilizing a quasi-experimental implementation science approach. This quality improvement project was deemed exempt by the Institutional Review Board of the hospital.

### 2.2. Setting and Study Participants

The hospital is the largest hospital in the region (314 beds) and the only hospital serving the county, which spans over 600 square miles, is home to approximately 180,000 residents, and has an 18% poverty rate. The hospital also provides care for residents of 11 surrounding counties. The criteria for service inclusion as multivisit patients were currently admitted patients with at least 2 admissions within the last year with at least 1 admission within the last six months. Participants had to be taking 1 or more chronic medications and be discharged back to the community setting to be eligible for the intervention.

### 2.3. Intervention

The TOC service was developed as a collaboration between the community hospital and a local college of pharmacy. The outpatient team was based at a pharmacist-led ambulatory care clinic housed within the college of pharmacy. External funding was awarded to pilot the model and provided for 1.0 full-time equivalent (FTE) pharmacist time, 0.15 FTE pharmacy resident time, 0.05 FTE outpatient social worker time, and 0.12 FTE program manager time to support the implementation and evaluation of the service. One ambulatory care pharmacist from the college of pharmacy clinic was embedded within the hospital 2.5 days per week to identify, recruit, and enroll eligible participants. The pharmacist utilized admissions reports to identify eligible patients and received referrals from case management and other hospital care providers through the hospital electronic medical record (EMR) and provider communication tools. During the inpatient stay, the pharmacist identified DRP and social risks, provided medication, disease state, and discharge counseling, and completed discharge medication reconciliation and outpatient appointment scheduling as availability allowed. Barriers to care were identified using the Health Capability Questionnaire^®^, available as part of the hospital’s care transitions software. The questionnaire identifies potential care barriers in nine categories including social, housing, financial, transportation, caregiver need, meals, smoking, engagement, and literacy. If the pharmacist was unavailable, the discharge medication reconciliation and counseling were not completed. Enrollment and discharge notes were entered into the EMR and faxed to the patient’s primary care and specialty care provider(s). After discharge, the pharmacist provided outpatient care from the college of pharmacy clinic during the remaining 2.5 days of the week. The outpatient intervention components consisted of contacting the patient by phone on approximately days 3, 7, 12, and 28 post-discharge to assess and address DRPs, barriers to care, and adherence to the care plan. An outpatient social worker contacted patients by phone after discharge to evaluate and address identified social needs and provided follow up, as indicated, to address care barriers. Additional appointments were completed by the pharmacist and social worker, if needed, to address patient care needs.

In the first three months of the service, the outpatient pharmacist shadowed the hospital’s inpatient pharmacists, learned about the hospital workflow, trained on the hospital’s electronic medical record (EMR) system, and worked with the TOC team to develop and test service protocols. Primary areas of focus included the hospital discharge process and electronic location for placement of service documentation. One barrier the team could not overcome was the need for documentation in two separate systems—one for the hospital and the other for the clinic. Once the workflow and documentation processes were in place, the pharmacist and program manager met with hospital resource groups, including the outpatient infusion center, dialysis unit, cardiac rehabilitation, hospitalist team, medical resident program, and a community palliative care service, to identify opportunities for collaboration. The program manager and service pharmacist began attending hospital unit meetings, including hospitalist, pulmonary, and case management team meetings, which facilitated the integration of the TOC service into the hospital workflow. The pharmacist and social worker communicated and collaborated with outpatient physician practices, pharmacies, and community organizations to maintain care continuity and address social risks and gaps in care.

### 2.4. Data Collection

Data were collected in real time from patient and service records to maintain dashboards utilized for routine reporting to service stakeholders. The outcomes of individuals who agreed to service enrollment and completed at least one outpatient visit with the pharmacist (intervention group) were compared to those who enrolled but were unreachable in the outpatient setting (comparison group). Interventions were tracked through service-specific codes added to EMR documentation and via commercial intervention tracking software. The service workflow can be found in Figure 1.

### 2.5. Measures

Due to the variability in TOC program design in the literature [13], the RE-AIM implementation science framework was used to provide structure to the evaluation of the impact of program components and identification of need for and assessment of the impact of program modification [19].

Reach was evaluated as the number of eligible participants, percentage of eligible participants engaging in the service, and population characteristics. This category also included evaluation of the number of individuals and admissions where individuals were identified but were excluded from participating in the service, the number where service participation was declined and the number and proportion of participants that were unreachable in the outpatient setting.

Effectiveness outcomes included assessment of the rate of 30-day and 90-day all-cause readmission and a comparison of readmission rates for intervention versus comparison group participants. Demographics for the two groups were compared using chi-square and Fisher’s exact tests for categorical variables and the t-test for continuous variables. An *a priori* alpha value of 0.05 was used for all analyses.

Adoption outcomes were not assessed. Due to having only a single pharmacist seeing patients, the service limited its engagement to a strategic group of providers including case management and a small group of physicians.

Implementation outcomes assessed how consistently the intervention was delivered including adherence to the service protocol, the adaptations made during implementation, and impact of those adaptations.

Maintenance outcomes included evaluation of long-term program adaptations persisting after the conclusion of grant funding. An anonymous survey was distributed to hospital providers (case managers, nurses, physicians) that had contact with the service 12 months after service initiation to assess their experience with the service and identify perceptions that could affect service sustainability.

## 3. Results

### 3.1. Reach

The study participation details are outlined in Figure 1. From February 2022 to August 2023, a total of 306 patients, accounting for 441 total admissions, were assessed for eligibility. Of those admissions, 234 (53%) were ineligible due to not meeting multivisit criteria or being transferred or discharged into the care of another medical facility/service. TOC services were declined by the patient or caregiver for 56 admissions, though some patients declining services in a prior admission agreed to utilize the TOC service during a subsequent admission. In 15% of admissions, the TOC pharmacist was not able to make contact with the patient during the admission and was, therefore, unable to recruit the individual to participate in the intervention. There were 84 eligible admissions where the patient agreed to participate, resulting in a final evaluated population of 66 patients due to 18 patients becoming ineligible after service enrollment. This ineligibility was due to the transfer of care to another facility or service. The participants had an average age of 60.1 ± 13.8 years, with 3.3 ± 1.8 admissions and 2.8 ± 4.1 emergency department (ED) visits in the year prior to the admission of service enrollment. There was an even split of male and female patients (50.0%) and the majority were black (66.7%) and had public insurance (78.8%). An average of 5.1 ± 1.8 barriers to care were identified during patient enrollment. There were no significant difference in characteristics for the comparison versus intervention groups (*p* >0.05). Additional demographics can be found in Table 1.

### 3.2. Effectiveness

The outcomes from only the index admission (admission of service enrollment) were included. Sixty-six participants were eligible to receive the service, 72% of which completed at least one outpatient visit with the pharmacist (Figure 1). There were no significant differences in characteristics between individuals enrolling in the service and completing no (comparison) versus at least one outpatient visit (intervention; *p* > 0.05; Table 1). Thirty-day readmission was 55.6% for the comparison group and 22.9% for the intervention group (*p* = 0.01). There was a nonsignificant 10.3%-lower rate of 90-day readmission in the intervention group compared to the comparison group (*p* = 0.50). Not all participants had reached the 90-day post-discharge time point at the time of data analysis. The time to readmission was 42.9 ± 62.3 versus 63.7 ± 94.3 days (*p* = 0.39) for the comparison versus intervention, respectively (Table 2). No significant correlations between variables and readmission were found.

The pharmacist documented 2097 TOC service interventions during this timeframe. This included interventions completed while the participants were hospitalized (both comparison and intervention groups) and post-discharge (intervention). This also included interventions completed in subsequent admissions for the 11 individuals who were readmitted if they continued to receive care during the subsequent hospitalization(s) and/or post-discharge. Individuals receiving adherence and compliance interventions (e.g., counseling, medication organizer) were significantly less likely to be readmitted within 30 days (*p* < 0.01). Intervention frequency, by category, is reported in Table 3. No relationships between baseline characteristics and intervention numbers or types were found (*p* > 0.05). The social worker documented 182 interventions for the 35 patients that were able to be contacted.

### 3.3. Implementation

#### 3.3.1. Protocol Adherence

The average time to patient identification was 1.7 ± 1.5 days. The completion rate for discharge medication reconciliation and counseling was low at 36.4%. Two key factors significantly impacted completion. The first was the pharmacist’s schedule. Performing outpatient care 50% of the time kept the pharmacist from actively reviewing inpatient records on a daily basis. Patients discharging on an outpatient care day typically did not receive a discharge medication review or counseling. Secondly, the rolling discharge schedule at the hospital made it difficult for the pharmacist to know when the final medication list was available for review without manually checking for its completion. There were no relationships found between completion of the reconciliation and counseling and participation in outpatient visits, nor between completion and 30-day readmission (*p* > 0.05). In total, 100% of patients received an SDoH assessment from the pharmacist and 100% of patient visits were documented within both EMR systems. Adherence to this component was facilitated by making it part of the service enrollment process. Completion of the discharge note and dissemination to the patient’s provider(s) was high at 90.1%. The reason for not completing the discharge note was delay between the discharge and pharmacist awareness of the discharge due to weekend discharge and/or the pharmacist being in-clinic on the day of discharge. For patients without a discharge note, the discharge information was included in the first outpatient note. All notes were faxed to the provider(s) according to the service protocol.

A total of 141 visits were completed by the pharmacist. The average time to first outpatient contact was 5.4 ± 5.1 days, due in part to participants not answering the phone for the first post-discharge call. The intervention group participants completed an average of 2.1 ± 1.9 of the 4 scheduled visits. There was no association between number of visits completed and readmission. The social worker documented 42 completed visits for 35 patients. Difficulties in reaching patients combined with low workload allocation for the social worker limited patient contact.

#### 3.3.2. Adaptations

At implementation, patients who were not reached for their scheduled appointment had it automatically rescheduled to the next day of outpatient appointments. It was quickly evident that this would overload the call schedule without an enhanced ability to reach the patient. The team decided to only contact patients according to the original schedule, so that a patient missing the day-3 appointment, for example, would not be contacted again until the day-7 appointment unless the patient initiated contact between the two appointments. If the patient initiated contact, the pharmacist would complete this visit at that time or attempt to contact them at the earliest opportunity. Due to the pharmacist’s time being split between the hospital and clinic, post-discharge appointments were scheduled as close to the schedule as possible. Also, due to the pharmacist’s schedule, the pharmacist was not consistently available to complete discharge medication reconciliation and counseling. Interim analyses indicated that not completing the medication reconciliation and counseling did not impact the rate of 30-day readmission. To improve the utilization of the pharmacist’s time and increase service capacity, these components were removed from the service protocol.

### 3.4. Maintenance

Extramural funding concluded 2 years after service implementation. At the conclusion of the funding period, the research team provided service performance data to the hospital administration. Based on the positive outcome of 30-day readmission reduction, the hospital agreed to continue to fund this service by providing the support of a 1.0 FTE pharmacist and 0.045 FTE project manager. However, the hospital’s focus expanded to include additional high-risk patient populations including patients admitted for heart failure and chronic obstructive pulmonary disease (COPD). A total of 392 patients were assessed between January 2023 and December 2024, and 142 were included in the service. Fifty-six point three percent of patients participated in outpatient visits with a continued significant reduction in 30-day all-cause readmission, and 7153 pharmacist interventions were documented.

Informal feedback was sought throughout the study period, resulting in training and reporting sessions with hospital provider teams. Workflow adaptations, including the development of consults within the EMR and addition of the consult option to specific order sets facilitated sustainable integration into the hospital workflow. Formal feedback from hospital providers was sought to inform the need for additional service modifications. A total of 16 hospital providers completed the anonymous survey, including 2 pharmacists, 3 nurses, 8 social workers, 1 clinical exercise physiologist, and 2 physicians, the entirety of the hospital team collaborating with the service. Overall, the survey respondents reported positive interactions with the service and impacts on patient care. Eighteen percent reported the service caused some disruption to the normal workflow. A total of 100% of providers reported that their experience with the TOC service and service team was either excellent (81% and 88%, respectively) or good (19% and 12%, respectively) (Table 4).

## 4. Discussion

Utilization of the RE-AIM framework for the evaluation of service implementation uncovered important features of the service that impact the feasibility and sustainability of this TOC service model. The study findings demonstrate the ability of the service to reach the target population: patients at increased risk for early hospital readmission due to having multiple recent admissions and the large number of missed opportunities due to the 50:50 split of pharmacist time between the hospital and clinic settings. The percentage of admissions meeting the eligibility criteria was approximately 47%. Multivisit patients likely had more advanced disease underlying their multiple admissions and a high percentage were discharged to rehab facilities, skilled nursing, or end-of-life care. This approach was effective at identifying a high-risk population, making this criterium potentially useful in prioritizing patients identified on the basis of disease state or other criteria. Barriers to care were prevalent, highlighting the prominence of social risks in the study population and the potential role of social work in care transitions in this population (reach). Since the end of the study period, the enrollment process has been modified to allow for outpatient enrollment, increasing service reach. The inability to reach participants in the outpatient setting has been addressed by utilizing automated and manual text messaging approaches, but it has yet to be overcome.

The results demonstrate the target population has a higher readmission risk than is reported for many of the high-risk disease states [3] and illustrate the effectiveness of the intervention in reducing that risk, lending support to implementation of the service model in similar settings. Though there was a trend in positive impact on 90-day readmission, a significant relationship was not found, potentially due to the small evaluated population or to the focus on the 30 day post-discharge period. Additional service modifications and/or hospital support to enhance care during the 60 days following the current service period could further impact 90-day readmission, though this would decrease available resources. The number and types of interventions performed highlight the extensive need for pharmacy and social services in the immediate post-discharge services for this patient population (effectiveness).

Due to time allocation and the clinical workflow, adherence to specific protocol aspects was not feasible. Fortunately, data analyses show that the modification of discharge activities and timing of outpatient visits did not significantly impact readmission rates. The ability of the service to identify and evaluate needed changes likely enhanced the success of the service due to its ability to adapt to the local context, a feature recommended for successful TOC service integration [20]. The social worker had limited ability to engage with patients. This could be overcome by increasing that allocation and potentially having shared visits with the pharmacist. Previous studies have established a linkage between social risks and increasing readmission risk, necessitating further integration of strategies for social barriers [21] (implementation).

The findings from this project have informed continuous service modification to optimize both efficiency and outcomes. The expansion of service inclusion criteria to align with hospital priorities and integration of the service into the hospital workflow have fostered its sustainability. The evaluation of hospital provider perceptions further supports sustainability, as the service collaborators have a positive view on both service activities and potential future impacts (maintenance).

There are several limitations to this study, including a small sample size and lack of randomization. Also, this work was completed in a single regional hospital, potentially reducing generalizability. Though the characteristics of the intervention and comparison group participants were compared, there is a risk of self-selection bias into each of these groups that could contribute to the differences in readmission outcomes.

Future randomized, controlled trials including a larger patient cohort, multiple sites, and enhanced integration of social workers should be undertaken to validate the results of this study and address the identified limitations. Future studies designed to assess the Adoption domain of the RE-AIM framework would be useful for translation of these findings into other populations and settings.

## 5. Conclusions

An ambulatory care-based pharmacist-led TOC service collaboration between a regional hospital and college of pharmacy clinic was successfully implemented based on a structured evaluation using the RE-AIM framework. Use of the framework allowed for understanding of the program features impacting service feasibility and sustainability. Importantly, this work provides evidence for the use of admission history as a criterion for identification of individuals at high risk for readmission and reinforces the value of pharmacists in care transitions.

## Figures and Tables

**Figure 1 pharmacy-13-00099-f001:**
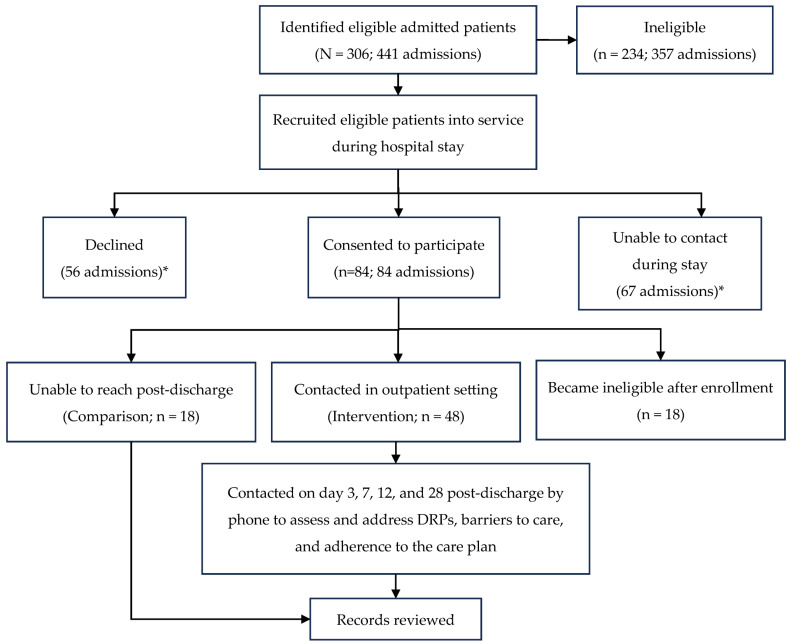
Transitions-of-care service workflow and participation. * Not all unique individuals.

**Table 1 pharmacy-13-00099-t001:** Participant characteristics.

Characteristics	Total (n = 66)	Comparison (n = 18)	Intervention (n = 48)	*p* Value ^1^
**Age, y, mean (SD)**	60.1 (13.8)	55.9 (14.5)	61.6 (13.3)	0.14
**Sex**				0.58
-Male, n (%)	33 (50.0%)	8 (55.6%)	25 (52.1%)	
-Female, n (%)	33 (50.0%)	10 (44.4%)	23 (47.9%)	
**Race**				0.56
-Black, n (%)	44 (66.7%)	11 (61.1%)	33 (68.8%)	
-White, n (%)	22 (33.3%)	7 (38.9%)	15 (31.3%)	
**Insurance status**				0.15
-Public, n (%)	52 (78.8%)	14 (77.8%)	38 (79.2%)	
-Private, n (%)	9 (13.6%)	1 (5.6%)	8 (16.7%)	
-Uninsured, n (%)	5 (7.6%)	3 (16.7%)	2 (4.1%)	
**LOS, days, mean (SD)**	6.4 (3.8)	6.2 (5.1)	6.5 (3.2)	0.82
**Annual admissions, mean (SD)**	3.3 (1.8)	3.7 (1.6)	3.1 (1.8)	0.20
**Annual ED admissions, mean (SD)**	2.8 (4.1)	4.3 (6.6)	2.2 (2.5)	0.20
**Number of diagnoses, mean (SD)**	10.3 (3.8)	10.3 (4.2)	10.2 (3.8)	0.92
**Number of barriers, mean (SD)**	5.1 (1.8)	4.8 (2.0)	5.3 (1.7)	0.34

^1^ Chi-square and Fisher’s exact tests for categorical variables and *t*-test for continuous variables. ED = emergency department; LOS = length of stay.

**Table 2 pharmacy-13-00099-t002:** Readmission comparisons.

	Total n = 66	Comparison n = 18	Intervention n = 48	*p* Value
30-day readmission, % (n)	31.8% (21)	55.6% (10)	22.9% (11)	0.01
90-day readmission, % (n) ^1^	49.2% (30)	57.1% (8)	46.8% (22)	0.50
Time to readmission, days, mean (SD)	58.0 (86.8)	42.9 (62.3)	63.7 (94.3)	0.39

^1^ Limited to 61 patients (comparison n = 14; intervention n = 47) due to some patients not reaching 90 days post-discharge at the time of data collection.

**Table 3 pharmacy-13-00099-t003:** Documented pharmacist and social worker interventions.

Primary Intervention	Number	Percentage
**Pharmacist Interventions**		
Patient assistance program/Medicare Part D	1	0.0%
Drug therapy initiated	1	0.0%
Vaccination administered	2	0.1%
Telephone assessment	2	0.1%
Drug therapy discontinued	3	0.1%
Therapeutic interchange performed	4	0.1%
Outpatient—patient home visit	4	0.1%
Therapeutic duplication avoided	5	0.2%
Therapeutic interchange recommended	5	0.2%
Inpatient—anticoagulation consult/follow-up	9	0.3%
Drug interaction (disease/drug/food/lab)	12	0.4%
Patient referral	19	0.6%
Drug therapy adjusted (dose/frequency/etc.)	19	0.6%
DME/medical devices	21	0.7%
ADR prevented	22	0.7%
Renal dose evaluation	24	0.7%
Outpatient—additional 15 min	33	1.0%
Vaccine recommended	35	1.1%
Clarification of orders	39	1.2%
Allergy info clarified	41	1.3%
OTC recommendation	45	1.4%
Outpatient—pharmacy care initial	52	1.6%
Medication reconciliation	58	1.8%
Patient medication history	75	2.3%
Outpatient—pharmacy care reassess	76	2.4%
Patient counseling—Brief	86	2.7%
Lab evaluation	95	2.9%
Patient counseling—extended	95	2.9%
Telephone assessment	126	3.9%
Continuity of care	145	4.5%
Chart review (outpatient)	200	6.2%
Inpatient encounter (chart review/rounding)	1870	58.0%
Pharmacist Total	3224	100%
**Social Worker Interventions**		
Identified mental health/substance abuse	1	1%
Referral to care network	1	1%
Community resident utility assistance education provided	1	1%
Referral to aging waiver services	1	1%
Identified low literacy	2	1%
Community resident food banks education provided	2	1%
Food stamps education provided	2	1%
Assessment of health/behavior subsequent visit	2	1%
Community resident transportation education provided	3	2%
Identified transportation issues	4	2%
Group health/behavior intervention	5	3%
Social Security benefits application education provided	7	4%
Medication assistance program education provided	8	4%
Identified limited financial resources	10	5%
Identified medication affordability issue	11	6%
Identified limited access to community resources	13	7%
Identified limited support system	15	8%
Provided community resources	26	14%
Individual health/behavior intervention	31	17%
Assessment of health/behavior initial	37	20%
Social Worker Total	182	100%
**Total Interventions**	3406	

DME = durable medical equipment; ADR = adverse drug reaction; OTC = over-the-counter.

**Table 4 pharmacy-13-00099-t004:** Hospital provider service perceptions.

Survey Question	Responses n (%)
The transitions of care (TOC) team works effectively.	
-Strongly agree	11 (69)
-Agree	5 (31)
-Disagree	0
-Strongly disagree	0
2. The TOC service runs smoothly.	
-Strongly agree	12 (75)
-Agree	4 (25)
-Disagree	0
-Strongly disagree	0
3. The TOC service disrupts my normal workflow.	
-Strongly agree	2 (13)
-Agree	1 (6)
-Disagree	4 (25)
-Strongly disagree	9 (56)
4. The TOC service aligns with my goals for patient care.	
-Strongly agree	13 (81)
-Agree	3 (19)
-Disagree	0
-Strongly disagree	0
5. I understand how the TOC service operates.	
-Strongly agree	10 (63)
-Agree	6 (38)
-Disagree	0
-Strongly disagree	0
6. The TOC service saves time and money for both myself and the hospital.	
-Strongly agree	10 (63)
-Agree	6 (38)
-Disagree	0
-Strongly disagree	0
7. The TOC service reduces readmission rates.	
-Strongly agree	10 (63)
-Agree	6 (38)
-Disagree	0
-Strongly disagree	0
8. The TOC service improves patient care and understanding.	
-Strongly agree	11 (69)
-Agree	5 (31)
-Disagree	0
-Strongly disagree	0
9. The TOC service has benefited my patients.	
-Strongly agree	10 (63)
-Agree	6 (38)
-Disagree	0
-Strongly disagree	0
10. I am confident that I can continue to effectively work with the TOC team.	
-Strongly agree	12 (75)
-Agree	4 (25)
-Disagree	0
-Strongly disagree	0

TOC = transitions of care.

## Data Availability

The raw data supporting the conclusions of this article will be made available by the authors on request.

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
