# Peer review of "The Use of RE-AIM to Evaluate a Pharmacist-Led Transitions of Care Service for Multivisit Patients at a Regional Hospital"

_pharmacy, 2025, doi:10.3390/pharmacy13040099_

Round 1

Reviewer 1 Report

Comments and Suggestions for Authors

The authors described the results of a study on pharmacist-led transitions of care services, and their findings are essential for health practice and the pharmacist's role in the healthcare system. The study was well-designed, and the results were presented clearly and adequately.

Below, I add a few comments that could improve the paper and make it more readable.

1) The authors may briefly describe the RE-AIM method in the introduction and justify its use in their study. Not all readers may be familiar with the Re-AIM method.

2) Some abbreviations were not expanded: FTE, Re-AIM, LOS (table 2)

3) Vers 168-171: total of 306 patients, accounting for 441 total admissions, were assessed for eligibility. Of those admissions, 234 (53%) we ineligible due to not meeting multivisit criteria or being transferred or discharged into the care of another medical facility/service. In the figure, one box labelled "Ineligible" shows n=244, while in the text, it is n=234. Please add the number of patients and admissions in the box called: "Identified eligible admitted patients".

I suppose that in verse 170, it should be 234 (53%) were ineligible, instead of "we ineligible" (language mistake).

4) Table 2. Is it possible to add the information about the mean number of days between the last and current hospital admission, or the number of patients included who were admitted within the previous 90 days before the current admission? In the inclusion criteria, it was the admission in the last six months. Especially among effectiveness indicators, authors focus on the 30-day and 90-day readmission rates.  

Author Response

Comments 1: The authors may briefly describe the RE-AIM method in the introduction and justify its use in their study. Not all readers may be familiar with the Re-AIM method.

Response 1: We agree with the need to provide additional background information on the RE-AIM framework for the audience. Two additional sentences have been added to the introduction (page 2, lines 70-74).

Comments 2: Some abbreviations were not expanded: FTE, Re-AIM, LOS (table 2)

Response 2: Agree. We have defined FTE in the text (p.3, line 96) and created a legend for tables 1 (p.6, line 198) and 2 (p.8, line 232).

Comments 3: Vers 168-171: total of 306 patients, accounting for 441 total admissions, were assessed for eligibility. Of those admissions, 234 (53%) we ineligible due to not meeting multivisit criteria or being transferred or discharged into the care of another medical facility/service. In the figure, one box labelled "Ineligible" shows n=244, while in the text, it is n=234.

Response 3: Thank you for identifying this typo. It has been updated to reflect 234 ineligible individuals.

Comments 4: Please add the number of patients and admissions in the box called: "Identified eligible admitted patients".

Response 4: Agree. This is an excellent recommendation. Figure 1 has been updated to include the number of potentially eligible admissions and patients.

Comments 5: I suppose that in verse 170, it should be 234 (53%) were ineligible, instead of "we ineligible" (language mistake).

Response 5: Thank you for catching this mistake. It has been corrected. (line 180)

Comments 6: Table 2. Is it possible to add the information about the mean number of days between the last and current hospital admission, or the number of patients included who were admitted within the previous 90 days before the current admission? In the inclusion criteria, it was the admission in the last six months. Especially among effectiveness indicators, authors focus on the 30-day and 90-day readmission rates. 

Response 6: Thank you for this suggestion. We had not considered this as a data point and recognize the potential impact on readmission even after participation in the service. We will assess whether to pursue this for future our future work.

Reviewer 2 Report

Comments and Suggestions for Authors

This study addresses a highly relevant topic, especially in the context of the growing number of patients with chronic conditions. The chosen methods (namely, the RE-AIM framework and quasi-experimental design) are appropriate and well-aligned with the study's objectives. The conclusions are supported by the results and demonstrate real-world applicability of the pharmacist-led transitions of care service.

Although the study has some limitations (such as a relatively small sample size, lack of randomization, and absence of an evaluation of the Adoption component) these are clearly acknowledged by the authors. Despite these constraints, the work is undoubtedly of interest and deserves publication following minor revisions:

  1. Improve the organization of the Methods and Results sections by introducing clear subheadings.

  2. Convert some of the result tables into visual diagrams to enhance readability and data presentation.

  3. Review table titles (particularly Table 2) to ensure they clearly reflect the content and purpose of the table.

  4. In the abstract, remove the use of all caps for the word "PURPOSE" and rephrase the sentence accordingly.

  5. Add a concluding paragraph addressing the potential for a future randomized controlled trial (RCT) using a similar design, with a larger cohort of patients, more pharmacists and social workers involved, and including an assessment of the Adoption domain.

After the minor revision paper can be accepted.

Author Response

Comments 1: Improve the organization of the Methods and Results sections by introducing clear subheadings.

Response 1: Thank you for this feedback. The methods and results are both divided by subheadings. It is possible that the formatting keeps the subheadings from standing out. We have made the adjustment of italicizing and bolding the subheadings within section and defer to the editor for additional methods for clearly subdividing these sections.

Comments 2: Convert some of the result tables into visual diagrams to enhance readability and data presentation.

Response 2: Thank you for this feedback. During preparation of the original submission, the authors reviewed multiple ways to format the data and had difficulty translating the volume of data, specifically for tables 3 and 4 into figures. We are open to additional suggestions of how to capture this data in more visual formats.

Comments 3: Review table titles (particularly Table 2) to ensure they clearly reflect the content and purpose of the table.

Response 3: Agree. We have updated the titles of Tables 2 and 3 to reflect the included content.

Comments 4: In the abstract, remove the use of all caps for the word "PURPOSE" and rephrase the sentence accordingly.

 Response 4: We agree and have updated the abstract according to the suggestion given.

Comments 5: Add a concluding paragraph addressing the potential for a future randomized controlled trial (RCT) using a similar design, with a larger cohort of patients, more pharmacists and social workers involved, and including an assessment of the Adoption domain.

Response 5: Agree. We have included an additional paragraph in the discussion section to address this feedback. (lines 361-365)

Reviewer 3 Report

Comments and Suggestions for Authors

The title of the manuscript is consistent with the content of the manuscript.

The abstract summarizes well the review.

In the Introduction section general information about the importance of the theme is presented.

Row 89: the abbreviation FTE should be explained

In the Materials and methods section the research procedure is described in detail. The project is well documented and designed.

The RE-AIM framework and/or concept can be shortly presented (where does the acronym comes from).

Row 170: “were ineligible” instead of “we ineligible”

The Results section clearly presents the results obtained.

Table 1, header: the total number of 66 is the sum of 48 Contacted in outpatient setting and 18 Unable to reach post-discharge?

Table 1: LOS abbreviation should be explained

Table 2: The table caption should be in accordance with the table’s content.

Instead of the number first and in the bracket the corresponding percentage, written in reverse order, the more relevant information would come first

Table 3: should be mentioned that the procedure codes are in brackets

The data in Table 4 can be presented as bar-chart or column-chart to facilitate transparency.

Row 235: SDoH assessment – the abbreviation should be explained

In the discussion section data presented in the Results section are explained.

The novelty of this study should be highlighted. There are similar studies published (e.g. 10.1310/hpj5106-468), the results can be compared.

The References are appropriate.

Editing errors should be corrected (e.g. Row 47: “readmission[9]”,  row 52: “support,[10]” space between text and bracket.

Author Response

Comments 1: Row 89: the abbreviation FTE should be explained

Response 1: Agree. We have defined FTE in the text (p.3, line 96) and created a legend for tables 1 (p.6, line 198) and 2 (p.8, line 232).

Comments 2: The RE-AIM framework and/or concept can be shortly presented (where does the acronym comes from).

Response 2: We agree with the need to provide additional background information on the RE-AIM framework for the audience. Two additional sentences have been added to the introduction (page 2, lines 70-74). 

Comments 3: Row 170: “were ineligible” instead of “we ineligible”

Response 3: Thank you for catching this mistake. It has been corrected. (line 180)

Comments 4: Table 1, header: the total number of 66 is the sum of 48 Contacted in outpatient setting and 18 Unable to reach post-discharge?

Response 4: Yes, this is correct.

Comments 5: Table 1: LOS abbreviation should be explained

Response 5: This has been corrected.

Comments 6: Table 2: The table caption should be in accordance with the table’s content. Instead of the number first and in the bracket the corresponding percentage, written in reverse order, the more relevant information would come first

Response 6: The suggested change has been made in the text.

Comments 7: Table 3: should be mentioned that the procedure codes are in brackets

Response 7: Thank you for pointing this out. We have removed the procedure codes.

Comments 8: The data in Table 4 can be presented as bar-chart or column-chart to facilitate transparency.

Response 8: Thank you for this feedback. The authors reviewed the visualization as a figure and decided that the table presented the data more clearly in this format.

Comments 9: Row 235: SDoH assessment – the abbreviation should be explained

Response 9: The abbreviation SDoH is defined within the introduction. (line 50)

Comments 10: In the discussion section data presented in the Results section are explained.

Response 10: Thank you for this feedback. Are there specific sections/lines that you believe should be modified? 

Comments 11: The novelty of this study should be highlighted. There are similar studies published (e.g. 10.1310/hpj5106-468), the results can be compared.

Response 11: This is valuable feedback. The team sought to highlight the novelty of this study in the introduction (lines 48-55), discussion (lines 328-337), and conclusion (lines 371-373). An additional sentence was added to the introduction (lines 56-57). We aimed to define the characteristics of the study population to inform the audience of the comparability to other populations studied in the literature. Are there additional opportunities to highlight the novelty that should be addressed?

Comments 12: Editing errors should be corrected (e.g. Row 47: “readmission[9]”,  row 52: “support,[10]” space between text and bracket.

Response 12: Thank you for pointing out these errors. They have been corrected in the text.

Reviewer 4 Report

Comments and Suggestions for Authors

This is a quasi-experimental quality improvement study using RE-AIM methodology to determine the feasibility of a ambulatory care-based pharmacist-led transitions of care practice to reduce hospital readmissions at one regional hospital in East Alabama. The manscript is generally well-written and referenced. I have only several suggestions that would strengthen the presentation.

Key words could be expanded to include "patient readmission" and "quality improvement." "Care transitions" and "pharmacist services" are not MeSH terms. Consider using  "pharmacist" and "transitional care."

In the introduction, the case for using RE-AIM is not strong. What is it about prior research that led you to select this strategy?

In the methods section, you mention at line 155 that adoption outcomes were not assessed. However, in the discussion, you mention that the hospital chose to fund the practice after the pilot funding ran out and expanded it to include other high-re-admission-risk diagnoses. I would claim that this is a great victory for service adoption, and you should mention this as I would guess that service establishment and sustainability were goals as well.

Table 4 needs to be reformatted so that the statements are clearly separated from the answers. I would number the survey statements and not center all of them. You should indicate when and how the surveys were administered. Since the program was conducted over a 1-year period, it may have been beneficial to re-survey quaterly to show adoption by the target provider group. More physician respondents would have aided your case.

In the discussion at line 303, you mention that multivisit patients may have had more advanced disease. Perhaps. They may also have been multimorbid with polypharmacy. A table showing disease breakdown would be helpful to the reader as well as discharge primary diagnoses and medications prescribed. There is no reporting of the number of patients that died since they may have been in your unreachable, lost to follow up, control group. How else could you have identifed your contro group?

References are not in MDPI style.

Thank you for the opportunity to review your interesting study.

Author Response

Comments 1: Key words could be expanded to include "patient readmission" and "quality improvement." "Care transitions" and "pharmacist services" are not MeSH terms. Consider using  "pharmacist" and "transitional care."

Response 1: The keywords have been updated to MeSH terms only and include pharmacists, transition care, implementation science, quality improvement

Comments 2: In the introduction, the case for using RE-AIM is not strong. What is it about prior research that led you to select this strategy?

Response 2: We agree with the need to provide additional background information on the RE-AIM framework for the audience. Two additional sentences have been added to the introduction (page 2, lines 70-74).

Comments 3: In the methods section, you mention at line 155 that adoption outcomes were not assessed. However, in the discussion, you mention that the hospital chose to fund the practice after the pilot funding ran out and expanded it to include other high-re-admission-risk diagnoses. I would claim that this is a great victory for service adoption, and you should mention this as I would guess that service establishment and sustainability were goals as well.

Response 3: We agree that hospital funding of the program is a big win. We utilized the definition from Re-Aim.org to guide our outcomes which is, “The absolute number, proportion, and representativeness of settings and intervention agents (people who deliver the program) who are willing to initiate a program, and why. Note that adoption can have many (nested) levels- for example, staff under a supervisor under a clinic or school, under a system, within a community” and felt that utilizing the word “adoption” to describe the continued funding may be confusing to the reader.

Comments 4: Table 4 needs to be reformatted so that the statements are clearly separated from the answers. I would number the survey statements and not center all of them. You should indicate when and how the surveys were administered.

Response 4: We agree that the formatting of Table 4 is not ideal. We have left justified the table contents and utilized numbering and bulleting to distinguish questions/statements from their answer choices.

Comments 5: Since the program was conducted over a 1-year period, it may have been beneficial to re-survey quaterly to show adoption by the target provider group. More physician respondents would have aided your case.

Response 5: Thank you for this comment. Due to the service capacity, we were unable to invite more providers to participate, making our potential sample very low. This sample represents nearly the entire population of providers that had touchpoints with the service.

Comments 6: In the discussion at line 303, you mention that multivisit patients may have had more advanced disease. Perhaps. They may also have been multimorbid with polypharmacy. A table showing disease breakdown would be helpful to the reader as well as discharge primary diagnoses and medications prescribed. There is no reporting of the number of patients that died since they may have been in your unreachable, lost to follow up, control group. How else could you have identifed your contro group?

Response 6: We agree that these factors could impact our results. In table 1 we report the average number of discharge diagnoses (10.2 ± 3.8), though we do not delineate the most common diagnoses. This is a potential data point for future investigations. We did not collect medication information but, likewise, is a potential data point for future investigation. Thank you for the suggestion.

Comments 7: References are not in MDPI style.

Response 7: Within the instructions for authors on the journal website is the following statement: “Your references may be in any style, provided that you use the consistent formatting throughout. It is essential to include author(s) name(s), journal or book title, article or chapter title (where required), year of publication, volume and issue (where appropriate) and pagination. DOI numbers (Digital Object Identifier) are not mandatory but highly encouraged. The bibliography software package EndNote, Zotero, Mendeley, Reference Manager are recommended.” (https://www.mdpi.com/journal/pharmacy/instructions) Zotero was used to format all references and a format that included internal citation utilizing brackets was chosen. Upon further review of the author instructions, there are alternative, conflicting instructions. The authors request clarification for the preferred citation format.